# The Physicochemical Characterization and In Vitro Digestibility of Maple Sugar Sand and Downgraded Maple Syrups

**DOI:** 10.3390/foods12193528

**Published:** 2023-09-22

**Authors:** Gautier Decabooter, Claudie Aspirault, Marie Filteau, Ismail Fliss

**Affiliations:** 1Département de Science des Aliments, Faculté des Sciences de l’Agriculture et de l’Alimentation (FSAA), Université Laval, Québec City, QC G1V 0A6, Canada; gautier.decabooter.1@ulaval.ca (G.D.); claudie.aspirault.1@ulaval.ca (C.A.); marie.filteau@fsaa.ulaval.ca (M.F.); 2Institut sur la Nutrition et les Aliments Fonctionnels (INAF), Québec City, QC G1V 0A6, Canada; 3Institut de Biologie Intégrative et des Systèmes (IBIS), Québec City, QC G1V 0A6, Canada

**Keywords:** maple syrup, buddy, ropy, sugar sand, TIM-1, in vitro digestion

## Abstract

The maple syrup industry generates substandard syrups and sugar sand as by-products, which are underused. In this study, we conducted a comprehensive analysis of the physicochemical composition of these products to assess their potential for valorization. Using HPLC analysis, we measured sugar and organic acid content as well as total polyphenol content using the Folin–Ciocalteu method. Additionally, we evaluated the in vitro digestibility using the TIM-1 model. We showed that the composition of ropy and buddy downgraded syrups is comparable to that of standard maple syrup, whereas sugar sand’s composition is highly variable, with carbohydrate content ranging from 5.01 mg/g to 652.89 mg/g and polyphenol content ranging from 11.30 µg/g to 120.95 µg/g. In vitro bioaccessibility reached 70% of total sugars for all by-products. Organic acid bioaccessibility from sugar sand and syrup reached 76% and 109% relative to standard maple syrup, respectively. Polyphenol bioaccessibility exceeded 100% during digestion. This can be attributed to favorable extraction conditions, the breakdown of complex polyphenol forms and the food matrix. In conclusion, our study demonstrates that sugar sand and downgraded maple syrups exhibit digestibility comparable to that of standard maple syrup. Consequently, they hold potential as a source of polyphenols, sugar or organic acids for applications such as industrial fermentation or livestock feeds.

## 1. Introduction

Maple syrup is a natural sweetener obtained by concentrating sap collected from sugar maple (*Acer saccharum*) and related trees (*Acer rubrum* and *Acer nigrum*). Maple sap harvest takes place mostly during the spring season and is favored by temperature fluctuations between sub-zero at night and above zero at daytime [1]. This is called the maple sugar season and starts generally as winter ends and lasts for about 4 to 6 weeks [2].

The nutritional composition of maple sap and its concentrated product, maple syrup, is well documented. Maple syrup is composed mainly of sucrose but also of polysaccharides (inulin and dextran), simple sugars (glucose and fructose), organic acids (malic, succinic and fumaric), ureides, amino acids, vitamins (B2 and B3), minerals (calcium, potassium and magnesium) and phytochemicals (mainly polyphenols) [3,4]. Several of these compounds have been studied for their potential beneficial effects on health. According to several studies, polyphenols extracted from maple syrup may have anti-inflammatory, antimicrobial, antioxidant, antimutagenic and anti-neurodegenerative properties, as well as other beneficial properties [5,6,7,8,9,10,11,12].

Continuous innovation in maple syrup production has significantly increased the quantity and quality of the syrup produced and, therefore, the industry’s profit margin. In 2021, the Canadian maple syrup industry accounted for about 80% of the world’s maple syrup production, which represents around 150 million pounds [13]. Canada is thus the leading global producer of maple products, with 89% originating from Quebec, 4% from Ontario and 7% from New Brunswick [13]. To be commercially sold as a “Canada Grade A” maple syrup, the product must meet the following five conditions: (i) its soluble solid content is 66–68.9% based on a refractometer or hydrometer measurement at 20 °C; (ii) it has no traces of fermentation; (iii) it is of uniform color and free of sediment and any cloudiness or turbidity; (iv) its color class is determined in terms of light transmission; and (v) it possesses a maple flavor characteristic of its color class and is free of any objectionable odor or taste [14]. Canada Grade A maple syrups are categorized in one of four color classes based on light transmittance at 560 nm: light (>75%), amber (50–75%), dark (25–50%) and very dark (<25%) [14,15].

Substandard maple syrups are downgraded or destroyed. In 2021, 10.3% of syrups were substandard due to defects in color, clarity, texture or flavor. Due to a lack of market outlets or potential uses for these products, the inventory of downgraded syrup has increased over the years, reaching 21% in 2018 compared with 11% in 2010. This represents significant economic losses for producers and an increasing storage burden for the maple syrup industry [13]. In this study, we investigated two types of defective maple syrups known as “buddy” maple syrup and ropy maple syrup.

Maple syrup presenting a buddy flavor is produced mainly at the end of the season, before the budding of the maple trees [16]. A change in the chemical composition of the sap takes place, resulting in syrup with much higher concentrations of nitrogen compounds [17,18]. It has been hypothesized that an interplay between tree metabolism and microbial activity contributes to the development of the buddy syrup flavor [17,18]. The exact mechanisms are unknown, but the buddy flavor compounds appear to include alkyl-pyrazines, formed likely from amino acids and reducing sugars or from urea and ammonia during heat processing [18,19,20,21]. Other compounds associated with this off-flavor are dimethyl disulfide and dimethyl trisulfide [16].

Ropy maple syrup results from the conversion of sugars to exopolysaccharides including dextran, arabinogalactans and rhamnogalacturonans by endogenous microorganisms in maple sap [22,23]. This defective product is defined as a syrup that can be drawn into filaments of 10 cm or more with a stick [17,24,25].

In addition to these substandard products, the maple industry generates a significant amount of by-product called sugar sand, which comes from the syrup filtration step that removes suspended solids [26]. There is currently no obvious use for this by-product. Quantification of sugar sand is difficult because its amounts vary depending on the sugar bush, the season and the day, even for the same production site [21,27]. Sugar sand is currently discarded as waste, which is a potential missed opportunity for the producer, since it could contain valuable nutrients.

The objectives of this study were (1) to determine the physicochemical characteristics of maple syrup by-products and (2) to investigate their digestibility using the TIM-1 digestive tract dynamic simulator. Based on their nutrient content alone, these by-products should find valuable uses in applications other than human nutrition.

## 2. Materials and Methods

### 2.1. Maple By-Product Samples

Three types of products derived from maple syrup production were evaluated: ropy maple syrup, buddy maple syrup and sugar sand. The bulk downgraded maple syrups were obtained from the Producteurs et Productrices Acéricoles du Québec (PPAQ, Quebec, QC, Canada) storage facilities following mandated inspection and classification. Three samples of ropy maple syrup were selected from various producers across Quebec, including two from Chaudière-Appalaches and one from Centre-du-Québec, all harvested during the 2017 season. Two composite buddy maple syrups were created by blending buddy syrups from around ten Quebec producers during the 2015 harvest. Standard commercial-grade pure maple syrup (Canada category A, dark, robust taste) was purchased from a local retailer in Quebec City. Sugar sand was obtained from five maple producers in the Lower St. Lawrence region, 2018 harvest season. All sugar sand samples were freeze-dried using an RePP model FFD-42-WS (Virtis, Gardiner, New York, NY, USA) to reduce moisture and thereby preserve product characteristics. Solid samples were also ground to fine powder in an A-11 basic analytical mill (IKA) for further analysis.

### 2.2. Carbohydrate Analysis

For sugar sand, sugars were extracted before analysis. Briefly, 2 g of sample was vortexed for 5 min with 10 mL of ethanol diluted to 80% vol. with nanopure water. The mixture was heated for 10 min at 70 °C in a water bath with constant agitation, vortexed again for 20 s and then centrifuged at 1000× *g* for 5 min. The supernatant was dried under reduced pressure in a rotavapor apparatus. The total sugar content was determined using the phenol–sulfuric acid method [28]. 

Sugars extracted from sugar sand and sugars in maple syrup samples were identified using a Waters Sugar-Pak I column (300 × 6.5 mm, 10 μm) on an Agilent 1100 series liquid chromatograph (Agilent Technologies Inc., Santa Clara, CA, USA) equipped with a refractive index detector (Agilent, 1260 Infinity). The isocratic solvent system contained calcium disodium EDTA (50 mg/L). Injected sample volume was 50 μL, the flow rate was 0.5 mL/min for a sample run time of 30 min and the column temperature was set at 90 °C.

### 2.3. Organic Acid Extraction and Analysis

For sugar sand, sugars were extracted before analysis by mixing 1 g of sample with 2 mL of ethanol (diluted as described above) for 5 min in an ultrasound bath and centrifugating it at 1000× *g* for 5 min. The pellet was extracted using the same ultrasound technique as above, and the supernatants were combined and concentrated to dryness under a nitrogen stream at 40 °C and then redissolved in 10 mL of water. The aqueous solution thus obtained was loaded into a Supelclean™ LC-18 SPE Tube (45 µm particle size, 60 Å pore size; 1 g sorbent mass, 6 mL reservoir volume, Sigma Aldrich, Saint Louis, MO, USA) preconditioned with methanol (5 mL), then water (5 mL), then 50:50 acetonitrile/water (10 mL) and vacuum-dried for 10 min. The sample (10 mL) was eluted with water, the first 5 mL was discarded and the next 5 mL was collected and diluted 1:5 in water for analysis via HPLC. All water was nanopure.

Malic, citric and fumaric acids were quantified using an HPLC method described elsewhere [29] with an Agilent 1100 series liquid chromatograph (Agilent Technologies Inc., Santa Clara, CA, USA) equipped with a Synergi Hydro-RP column (300 × 6.5 mm, 10 μm) and a UV detector. The isocratic solvent was 0.2 M KH_2_PO_4_, with the pH adjusted to 2.4, with phosphoric acid. The injected sample volume was 10 μL and the elution rate was 0.8 mL/min at ambient temperature. The needle was cleaned with solvent after each injection. Calibration curves and retention times of standards (Sigma Aldrich) were used to identify and quantify the organic acids, based on absorption at 214 nm.

### 2.4. Mineral Composition

Calcium, iron, potassium, magnesium, manganese, sodium, phosphorus, lead and zinc concentrations were determined through inductively coupled plasma optical emission using an Agilent 5110 SVDV ICP-OES spectroscope (Agilent Technologies, Victoria, Australia) at the following wavelength (in nm) pairs: 317/933, 393/366, 396/847 and 422/673 for Ca; 234/350, 238/204, 239/563 and 259/940 for Fe; 766/491 and 769/897 for K; 279/553, 279/800, 280/270 and 285/213 for Mg; 257/610, 259/372, 260/568 and 294/921 for Mn; 588/995 and 589/592 for Na; 177/434, 178/222, 213/618 and 214/914 for P; 182/143, 217/000, 220/353 and 283/305 for Pb; 202/548, 206/200, 213/857 and 334/502 for Zn; 371/029 for the Y internal standard. All ion analyses were performed in axial and/or radial view and in triplicate. Results are expressed in mg/g except for lead (µg/g).

### 2.5. Total Polyphenol Content

Total polyphenol content was determined using a Folin–Ciocalteu assay [30], modified for use in 96-well microplates [31] with slight modifications. The first three columns were loaded with 0, 10, 20, 30, 40, 50, 60 or 70 µL of a gallic acid standard (50 µg/mL) and then adjusted accordingly with nanopure water to obtain a total volume of 180 µL. The sample aliquot volume was 20 µL, also adjusted to 180 µL with nanopure water. Folin–Ciocalteu reagent solution (20 µL) was added to each well, and the plate was held in the dark for 10 min, then 50 µL of sodium carbonate was added to each well, followed by 20 min in the dark. The procedure was conducted at room temperature. Absorbance was monitored at 760 nm and the results are expressed as gallic acid equivalents (μg/mL) mean ± standard deviation for at least three replicates.

### 2.6. Physicochemical Properties

Total soluble solids were measured at 20 °C as °Brix using a digital refractometer (AR200, Reichert). Viscosity was measured at 37 °C using an Ares-G2 rheometer (TA Instruments, Newcastle, DE, USA) with serrated 25 mm parallel-plate geometry and a 1.0 mm gap. Color was determined using a Shimadzu UV-2600 UV-vis spectrophotometer (Shimadzu Scientific Instruments, Kyoto, Japan) at 560 nm in the transmittance mode. Color was determined by measuring the specific color parameters, where L, a and b signify, respectively, the color brightness, red and yellow parameters. The syrups were graded according to Canadian standards. The pH was measured using a bench-top pH meter (B10P, VWR^®^sympHony™) with a refillable glass electrode calibrated at pH 4.00 and 7.00.

### 2.7. Evaluation of By-Product Digestibility

The TIM-1 model (The TIM Company, Zeist, The Netherlands) described by Minekus et al. [32] was adapted to porcine conditions [33] and used to assess the digestibility of the by-products under porcine gastrointestinal pH and enzymatic conditions. Gastric secretions consisted of 150 U/mg lipase (Amano Enzymes, Elgin, IL, USA) and 3200 U/mg pepsin (from porcine gastric mucosa Sigma Aldrich) in gastric electrolyte solution (NaCl 6.2 g/L, KCl 2.2 g/L, CaCl_2_ 0.3 g/L, NaHCO_3_ 1.5 g/L). Porcine bile extract (Sigma Aldrich) and 8×USP pancreatin (Sigma Aldrich) dissolved in small intestine electrolyte solution (NaCl 5.0 g/L, KCl 0.60 g/L, CaCl_2_ 0.30 g/L) were injected into the duodenal compartment. Hollow-fiber modules connected to the jejunal and ileal compartments allow dialysis against a small intestinal electrolyte solution to simulate absorption. The liquid meal consisted of 149 g of syrup and 149 g of deionized water. The meal from sugar sand was prepared as follows: 60 g was hydrated with 180 g of nanopure water, acidified at pH 3.0 with 1 M HCl and stirred for 5 h. The preparation was centrifugated for 10 min at 10,000× *g* and the supernatant was recovered. The pellet was washed with 120 mL of nanopure water before being centrifuged, and the supernatant was recovered. This step was repeated once again before combining the three supernatants to obtain the meal from sugar sand. This was carried out to accommodate the insolubility of sugar sand, which would cause mechanical failure. Meal sample, apparatus initial content and secretion solutions were deoxygenated using bubbling nitrogen gas for 60 s. Five minutes before injecting the meal into the gastric compartment, 2 mL of alpha-amylase (33 U/mL, aqueous solution, Sigma Aldrich) was added, and the meal was placed in a water bath at 37 °C. Aliquots (6 mL) were removed from the gastric compartment at 0, 30, 90 and 180 min, 60 and 180 min from the duodenal compartment and 60, 180 and 360 min from the jejunal and ileal compartments. All sampled volumes were compensated for by compartment-specific solutions, so as to not affect the digestion rate. Effluents were collected, weighed and aliquoted at 60, 120, 180, 240, 300 and 360 min. The chyme included the remaining suspension within the duodenal, jejunal and ileal compartments after the completion of digestion. Dialysis fluids from the jejunal and ileal compartments were collected, mixed and weighed each hour, and 50 mL aliquots was kept. All aliquots were stored at −20 °C and analyzed separately for their composition (sugar, organic acids and total polyphenol content) with methods described earlier, without the extraction step. All digestions were performed in duplicate.

### 2.8. Statistical Analysis

All assays were performed at least in duplicate. Data were reported as mean ± standard deviation. Significance of the results (*p* < 0.05) was determined using Welch’s test in JMP^®^, Version 16, SAS Institute Inc., Cary, NC, USA (©1989–2023).

## 3. Results and Discussion

### 3.1. Physicochemical Properties

Maple syrup color, pH, total soluble solids, viscosity and transmittance are shown in Table 1. In terms of total soluble solid content, the standard and ropy syrup samples are within the regulatory values, whereas both buddy syrup samples are slightly below the minimum (66%). Based on transmittance, all syrup samples would be categorized as “Dark, Robust Taste” in Canada’s maple syrup classification. In the CIELAB tristimulus color space system, L is a measurement of sample darkness, zero being black and 100 being white, the “*a*” value indicates redness when positive and greenness when negative, and the “*b*” value indicates yellowness when positive and blueness when negative. *C*ab* quantifies color intensity and saturation, with higher values signifying vibrant hues and lower values denoting desaturated colors. The L values ranged from 62.87 to 19.17, with the highest being “Buddy-1” and the lowest being “Ropy-2”. The syrup color was a mixture of yellow and red with positive values of *a* and *b* values for the six samples, ranging from 15.36 to 36.45 and 32.34 to 92.16, respectively. The saturation value *C*ab* was the lowest for “Ropy-2” at 40.77, indicating a less vivid color compared with the highest saturation, which was 98.06 for the standard syrup.

The measured values are in agreement with the visual perception of dark syrup with a yellow orange hue/chroma and fall within reported values for maple syrup [18]. Maple syrup’s color is acquired during heating and evaporation of the sap and is attributed to a combination of caramelization and Maillard reactions [21], but it has been hypothesized that the thermal degradation of some compounds could also contribute, in particular to its redness, which increases when the sap is collected in late season close to bud break [18].

The pH of the syrups ranged from 5.80 to 6.84 with a mean of 6.36. Values below the typical range given in the literature (6.00–7.40) are probably due to microbial activity [34]. The pH of maple sap is usually in the 3.4−6.7 range, the variation being attributable to different degrees of conversion of organic acids to flavor compounds and/or microbial contamination [35].

The viscosity of the standard syrup and the buddy syrups were similar and well within the range reported in the literature (0.12–0.19 Pa·s) for syrups without viscosity defects [22,36]. As expected, the ropy syrups were more viscous, which is what defines this defect, presumably because of the presence of microbial exopolysaccharides [22].

### 3.2. Carbohydrate Content

The usual carbohydrate content of maple syrup is summarized in Appendix A. The sugar concentrations (sucrose, glucose and fructose) in the standard sample lie within the ranges mentioned in the literature: 42.30–75.60% for sucrose (average 64.18–66.00%), 0.00–9.60% for glucose (average 0.11–0.40%) and 0.00–6.80% for fructose (average 0.14–0.50) [27,36,37,38,39,40]. Ropy and buddy syrups all had sucrose concentrations that were at least 20% lower than the previously reported average for commercial maple syrup and above-average glucose and fructose concentrations (Table 1). This may be due to invertase or other microbial activity [36]. The monosaccharide levels in maple syrup are known to increase with storage time [37]. The processing method and/or microbial load may also contribute to this change [18,41].

The carbohydrate content of the sugar sands (Table 2, Figure 1 and published values in Appendix A) can vary by over 100-fold from one producer to another and nearly as much from a single producer, and it varies widely from one sap harvesting day to the next. Producers B and D, in particular, show great daily variability. This may be due to daily differences in processing (e.g., filter conditions, rinse water volume, etc.). Such variation has long been known (e.g., 4.40–85.22% according to [42]), suggesting that the yield of maple syrup could be increased by maximizing the desugaring of the filter cake (sugar sand).

HPLC analysis revealed that most sugar sands contain mainly sucrose (>92%), although one sample (B2) contained only 21% sucrose and 76% fructose. Since this particular sample was watery with a low sugar concentration, it is possible that contaminating microorganisms may have altered the sugar composition.

### 3.3. Organic Acids

Citric acid was not detected in the syrups (Table 1). The standard syrup contained malic, citric and fumaric acids within the ranges described in the literature, respectively, 0–0.90%, 0–0.047% and 0–0.130%. The ropy syrups also contained malic and fumaric acids at concentrations in these ranges, whereas one buddy syrup contained slightly more malic acid and the other close to the upper limit. The ropy and the buddy syrups also contained more fumaric acid in all cases. 

Citric acid was detected in the sugar sands, as were much higher concentrations of malic acid, and in most cases, at least slightly higher concentrations of fumaric acid were detected (Table 2). High concentrations of these organic acids in the filter cake may be associated with an off-flavor of the corresponding syrup [27].

### 3.4. Mineral Content

The mineral content of the maple products is summarized in Table 1 and Table 3. In the downgraded syrups, the total mineral concentration ranged from 2432 ppm to 3010 ppm, compared with 2852 ppm for standard syrup. At least one third of the mineral content was calcium, nearly two thirds in one ropy syrup, which is well over the range reported in the PPAQ industrial fact sheet [39]. Concentrations of iron, magnesium, manganese and lead all lie within the previously published ranges. However, potassium was below the range reported by Heiligman (541–4031 ppm with a mean of 2283 ppm) and the PPAQ (973–3960 ppm with a mean of 2404 ppm), whereas sodium was at least 10 times higher than the averages reported by Heiligman (36 ppm) and the PPAQ (14.4 ppm) and around or above the upper values (492 ppm and 90 ppm). Phosphorus was also above the range reported by Heiligman (the PPAQ did not quantify this mineral). And finally, zinc was slightly above the PPAQ value (Heiligman did not quantify this mineral).

For the mineral content analysis, the sugar sand samples were pooled according to saccharide concentration, low (less than 100 mg/g, see Table 2), intermediate (up to 300 mg/g) or high (over 400 mg/g). The total mineral concentrations (Table 3) were far higher than in syrup: 26-fold in the low-sugar pool, 40-fold in the intermediate pool and 22-fold in the high-sugar pool. Calcium was again the main mineral, representing at least 87% of the total mineral content and constituting over 10% of the by-product’s dried weight. The other minerals were present in much smaller amounts, which were, in decreasing order: magnesium (about 0.3%), phosphorus (about 0.23%), manganese (less than 0.1%), potassium, sodium, zinc, iron and less than 1 ppm of lead. Compared with the only published values we could find [43], the iron concentration was lower than the reported range (38–1250 ppm) and the magnesium concentration, at about 3000 ppm, was slightly higher (600–2900 ppm).

### 3.5. Total Polyphenol Content

Using the Folin–Ciocalteu method to measure polyphenol content allowed us to compare our results directly with published values for dark, robust taste syrups [39]. The total polyphenol contents in gallic acid equivalents were found to range from 133 to 339 µg per g, compared with 231 µg/g in the case of standard maple syrup. This is lower than the PPAQ averages of 1183 µg/g and a range of 488–2124 µg/g based on the analysis of 135 dark, robust taste syrups. Gallic acid equivalent values may reach 8040 µg/g for very dark maple syrup [31], nearly four times the maximum reported by the PPAQ. Such variations have been attributed to the well-known limitation of the Folin–Ciocalteu assay [44,45], which is an indirect method that can be influenced by non-phenolic compounds (e.g., sugars, organic acids) [46]. The Fast Blue BB method is more direct [40] and has reported 654.78 ± 166.70 GAE μg/mL for 13 samples of dark grade maple syrup, which converts to 494.42 μg/g based on an absolute chromatographic method [47]. Owing to differences in methods, at this time, direct comparisons of different studies of polyphenol content are difficult and qualitative at best.

The polyphenol content of the sugar sand was found to be almost an order of magnitude lower (11.30–120.95 µg GAE/g). This is, to the best of our knowledge, the first time the total polyphenol content of sugar sand has been reported. 

### 3.6. Evaluation of the Digestibility of Maple Syrup By-Products

In the TIM-1 digestion model, compounds collected in the jejunum and ileum dialysates are considered bioaccessible, and those collected in the chyme are considered potentially bioaccessible. The ileal effluent represents the compounds that would reach the colon and would thus be available to the high-density and diverse intestinal microbiota. The bioaccessibility of sugars, organic acids and polyphenols in maple by-products throughout the non-colonic digestive cycle is summarized in Figure 2.

At least 10% of the sugar, mainly the monosaccharides glucose and fructose, is absorbed within 60 min (Figure 2A). After 180 min, more than half the sugar has been absorbed. At the end of the digestion, 70% of the input sugar has been absorbed from ropy syrup and 86.80% has been absorbed from standard maple syrup. These levels would certainly have been higher with a complex meal (containing lipids) which tends to slow down gastrointestinal transit. It has been reported that polyphenols can inhibit digestive carbohydrases and appear to modulate glucose uptake from carbohydrate-rich meals through other mechanisms that remain to be demonstrated clearly [48]. However, for sugars at the end of the simulated digestion, no significant difference was observed between the SMS and the tested products (Dunnett’s test, *p* > 0.05).

Figure 2B shows two different patterns for organic acid absorption. Organic acids are absorbed completely from the syrups after 300 min—even by more than 100% relative to the input. These excess absorption values are probably due to differences between our extraction efficiency for by-product content analysis and extraction of the same compounds under the physicochemical conditions of the TIM-1, and hence, due to an underestimation of their actual concentrations in the samples. Since the pH in the gastric compartment is lower than the pKa of almost all carboxyl groups, the organic acid molecules are mostly protonated and therefore less inclined to complex with minerals. In addition, the temperature in the digestive tract (37 °C) may favor the release of organic acids. The 6 h digestion time versus 5 min laboratory extraction time may also be expected to increase organic acid solubilization, especially since the peristaltic movement likely provided more thorough mixing, and the higher liquid/solid ratio used in the simulator likely improved mass transfer. These parameters all need to be explored to determine if organic acid extraction during sample analysis is incomplete and needs improvement [49,50,51]. A more efficient treatment, feasible at large-scale such as hydrodynamic cavitation, may be necessary [52]. Finally, for organic acids, at the end of the simulated digestion, a significant difference was observed between the SMS and two samples: SMS-LS and SMS-MS (Dunnett’s test, *p* = 0.04), suggesting a lower nutritional value when the residual sugar content is low in sugar sand.

The absorption of polyphenols increased quickly, reaching nearly 100% of the input at 180 min for all products (Figure 2C). Moreover, it kept increasing until the end of digestion, reaching more than 300% for the low-saccharide sugar sand. Also, for polyphenols, no significant difference was observed at the end of the simulated digestion between the SMS and the tested products (Dunnett’s test, *p* > 0.05). Similar to what appears to have been the case for organic acid absorption, the conditions provided by the TIM-1 likely favored polyphenol extraction to such an extent that they revealed the inadequacy of the analytical extraction step. It has been found previously that lower pH, warmer temperature, longer mixing time, higher liquid/solid ratio and digestive secretions are all favorable conditions for the liberation of polyphenols [53,54,55,56,57,58,59]. In addition, although we assayed total polyphenols, our method likely did not detect high molecular mass polymerized forms, which apparently can be hydrolyzed into lower molecular mass compounds that are detectable. This phenomenon has been observed and described previously [60]. Some polyphenols are bound tightly to other compounds or otherwise complexed in less bioaccessible forms that may break up during digestion, resulting in higher levels of polyphenols available for absorption. Few studies have focused on polyphenol release from syrup or other maple sap products or from any whole food [61], making it difficult to compare our results with the literature. As shown here, whole food studies may give substantially different results.

Overall, the downgraded syrups and sugar sand with high sugar content are not significantly different from the standard maple syrup, suggesting a similar nutritional value. It must be kept in mind that this in vitro model mimics the porcine digestive system to a large extent, but not perfectly, since it lacks endothelial cells and has nowhere near the surface area of a real small intestine and therefore cannot reproduce real rates of nutrient absorption from chyme into the bloodstream and surrounding tissues. Nevertheless, it shows that maple syrup by-products may be absorbed very well in vivo.

## 4. Conclusions

Considering the wide differences between pre-2000 data and the 2018 PPAQ data, we strongly recommend that future studies of maple syrup refer to the PPAQ for product composition. Although earlier studies remain a goldmine of information and their pioneering work is invaluable, some average values apparently include syrups that would now be classified as defective. Furthermore, the Quebec maple syrup industry has developed to the point that the PPAQ has been able to establish strict quality standards and promote the use of technologies that were unavailable prior to 2000. The next step may be the development and the publication of standard methods for maple syrup analysis.

Defective syrups remain a good source of polyphenols, which are becoming popular nutraceutical components as knowledge of their beneficial properties spreads among consumers [62]. Polyphenols can be extracted from defective syrups using a variety of means without loss of their high-value biological properties [63,64]. Furthermore, the residue remains a good source of sugars for industrial uses such fermentation feedstock or ethanol production [65]. 

In view of the composition of sugar sand and its digestibility in the TIM-1, we believe that the filter cake could be washed with hot water to provide additional sugar and added to the maple sap in the evaporator or before the reverse osmosis step, resulting in a yield gain. The washed residue could be dried at minimal cost [66] and fed to livestock. As a source of organic acids, this by-product could be used to promote animal health and performance [67,68,69,70], which would contribute to reducing antibiotic use in livestock production [71].

Finally, this study shows that the maple industry’s by-products have several potential uses and suggests possible future applications for these materials.

## Figures and Tables

**Figure 1 foods-12-03528-f001:**
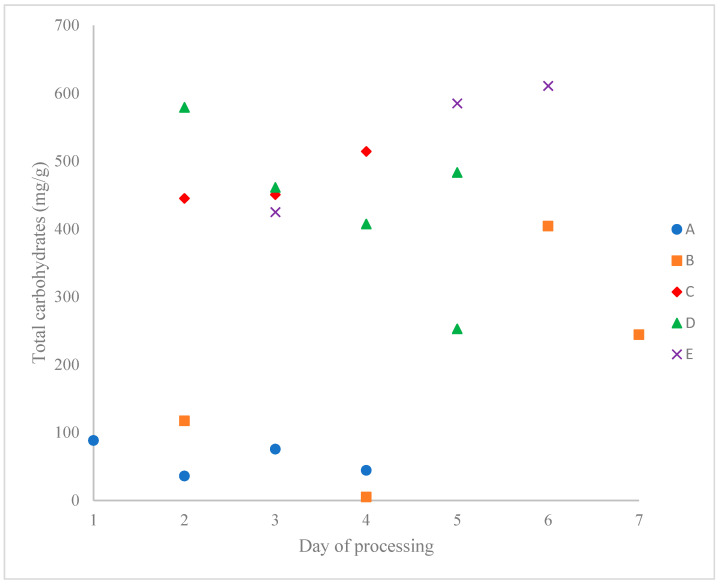
Carbohydrate contents of sugar sand reported according to the producer for different maple sap harvest dates (Producer A: blue circle; B: orange square; C: red diamond; D: green triangle and E: purple cross).

**Figure 2 foods-12-03528-f002:**
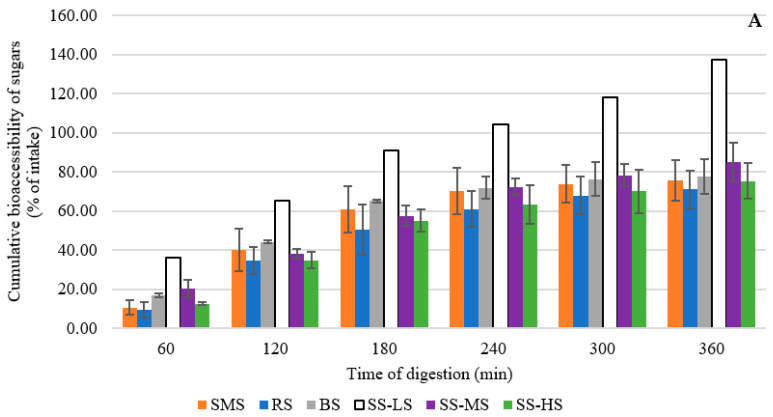
Bioaccessibility of sugars (**A**), organic acids (**B**) and polyphenols (**C**) in maple syrup and by-products, based on in vitro digestion (TIM-1), expressed as a percentage of the sample input (300 g). Error bars represent standard deviation. Empty bars indicate that only one value was obtained. SMS: standard maple syrup; SS-LS: sugar sand with low sugar content; SS-MS: sugar sand with intermediate sugar content; SS-HS: sugar sand with high sugar content.

**Table 1 foods-12-03528-t001:** Physicochemical analyses of maple syrups.

Analysis	Standard Syrup	Ropy-1	Ropy-2	Ropy-3	Buddy-1	Buddy-2
General						
Brix (%)	66.3	68.2	67.7	67.1	65.5	65.2
Viscosity (Pa·s)	0.16	0.52	0.75	0.49	0.17	0.15
pH	6.35	6.25	5.80	6.25	6.84	6.69
Transmittance (% at 560 nm)	30.60	30.47	47.51	44.75	30.60	26.47
Color analysis ^‡^						
L	56.89	58.34	19.17	30.91	62.87	47.89
*a*	33.50	26.28	24.82	36.45	30.81	15.36
*b*	92.16	70.43	32.34	52.55	82.03	51.97
*C*ab*	98.06	75.17	40.77	63.96	87.63	54.19
Carbohydrates (mg/g) ^†^						
Total carbohydrate	728.17 ± 29.01	601.45 ± 5.98	563.67 ± 5.55	609.94 ± 18.54	588.78 ± 31.54	551.48 ± 3.70
Sucrose	678.87 ± 5.21	533.25 ± 4.24	494.08 ± 2.21	544 ± 4.21	522.14 ± 3.65	382.3 ± 4.20
Glucose	8.9 ± 1.01	10 ± 0.87	24.17 ± 0.72	10.75 ± 1.10	23 ± 1.33	57.4 ± 2.22
Fructose	4.92 ± 0.05	8.3 ± 0.03	21.57 ± 1.20	10.8 ± 0.98	20.08 ± 1.51	31.2 ± 1.68
Organic acids (mg/g) ^†^						
Malic	6.12 ± 0.13	7.63 ± 0.18	7.01 ± 0.05	6.53 ± 0.31	9.13 ± 0.05	8.45 ± 0.24
Citric	n.d	n.d	n.d	n.d	n.d	n.d
Fumaric	0.1 ± 0.01	0.2 ± 0.01	0.22 ± 0.00	0.32 ± 0.01	0.15 ± 0.01	0.18 ± 0.00
Total phenolics (µg/g) ^†^	230.89 ± 1.29	189.93 ± 1.08	338.67 ± 9.94	315.62 ± 25.26	133.13 ± 3.78	149.38 ± 11.54
Minerals (ppm) ^†^						
Calcium	1144.88 ± 235.88	1938.50 ± 227.64	1379.53 ± 235.37	1488.33 ± 228.79	1271.58 ± 32.20	1215.90 ± 57.10
Iron	10.56 ± 3.98	5.45 ± 1.95	9.83 ± 3.80	8.01 ± 2.80	4.74 ± 0.66	9.45 ± 1.49
Potassium	522.32 ± 153.01	397.40 ± 219.69	435.41 ± 194.83	517.35 ± 181.42	420.26 ± 77.67	456.33 ± 148.20
Magnesium	230.93 ± 30.84	255.25 ± 13.21	243.97 ± 15.57	250.20 ± 15.07	207.23 ± 20.74	218.08 ± 28.34
Manganese	12.08 ± 5.84	34.43 ± 6.28	20.14 ± 6.96	23.28 ± 6.72	12.79 ± 3.52	13.49 ± 4.84
Sodium	551.41 ± 144.27	314.88 ± 201.26	341.62 ± 197.14	315.34 ± 184.60	478.62 ± 118.16	538.48 ± 144.17
Phosphorus	363.45 ± 607.13	46.48 ± 27.99	272.82 ± 541.10	64.88 ± 341.98	12.68 ± 0.73	343.69 ± 5.16
Zinc	16.15 ± 12.64	17.71 ± 7.71	17.20 ± 11.52	17.63 ± 9.35	24.00 ± 1.09	22.18 ± 2.31
Lead	0.16 ± 0.07	0.15 ± 0.09	0.11 ± 0.91	0.14 ± 0.08	0.20 ± 0.03	0.12 ± 0.03

^†^ Mean ± SD. n.d = not detected. ^‡^ CIELAB Color Space where L = Lightness; *a* = green to red gradient; *b* = blue to yellow gradient and *C*ab* = color intensity and saturation.

**Table 2 foods-12-03528-t002:** Chemical analyses of sugar sands *.

Sample	A1	A2	A3	A4	B1	B2	B3	B4	C1	C2	C3	D1	D2	D3	D4	D5	E1	E2	E3
Moisture (%)	66.73	70.54	67.35	70.14	57.77	57.06	57.08	75.21	39.19	39.8	47.27	41.32	39.935	65.56	51.85	53.236	54.87	35.30	25.32
Carbohydrates (mg/g)																			
Total	88.30 ± 0.95	36.03 ± 0.62	75.58 ± 2.08	44.36 ± 1.78	117.33 ± 12.18	5.01 ± 0.17	404.19 ± 26.81	244.26 ± 11.72	445.04 ± 3.68	450.66 ± 26.34	514.28 ± 9.44	579.28 ± 18.76	461.22 ± 17.31	407.50 ± 20.16	483.21 ± 16.75	252.84 ± 3.04	424.93 ± 5.48	585.18 ± 7.95	610.87 ± 11.52
Sucrose	89.49 ± 1.39	35.05 ± 2.22	64.21 ± 2.09	44.06 ± 1.88	95.63 ± 1.71	1.23 ± 0.17	393.52 ± 5.16	232.49 ± 4.30	405.85 ± 4.48	440.68 ± 5.43	468.72 ± 5.93	554.52 ± 7.13	448.13 ± 5.26	374.52 ± 4.58	442.69 ± 6.19	232.45 ± 4.24	405.86 ± 4.30	534.67 ± 5.71	570.80 ± 7.70
Glucose	n.d	n.d	4.00 ± 0.13	n.d	n.d	0.15 ± 0.02	n.d	n.d	n.d	n.d	6.98 ± 0.34	n.d	7.40 ± 0.04	6.93 ± 0.27	9.57 ± 0.91	5.09 ± 0.00	6.75 ± 0.57	n.d	n.d
Fructose	n.d	n.d	0.90 ± 0.04	n.d	n.d	4.29 ± 0.17	2.81 ± 0.23	2.12 ± 0.14	n.d	n.d	4.01 ± 0.11	n.d	5.22 ± 0.17	4.79 ± 0.37	6.67 ± 0.05	4.10 ± 0.01	4.33 ± 0.03	n.d	n.d
Organic acids (mg/g)																			
Malic	69.05 ± 1.89	71.12 ± 1.19	75.73 ± 2.71	80.72 ± 2.63	72.04 ± 1.46	98.87 ± 2.63	90.94 ± 1.91	93.14 ± 2.59	73.68 ± 1.80	77.38 ± 2.28	74.88 ± 1.06	75.84 ± 2.72	79.78 ± 2.69	88.78 ± 2.21	91.34 ± 1.78	103.86 ± 2.64	74.96 ± 2.71	67.89 ± 2.27	80.45 ± 2.26
Citric	1.20 ± 0.02	1.36 ± 0.11	1.75 ± 0.14	2.76 ± 0.17	1.56 ± 0.07	3.77 ± 0.18	2.97 ± 0.07	8.00 ± 0.28	1.36 ± 0.04	1.60 ± 0.05	1.66 ± 0.14	1.68 ± 0.14	1.81 ± 0.11	1.95 ± 0.16	2.33 ± 0.04	3.59 ± 0.28	1.66 ± 0.14	1.83 ± 0.18	3.04 ± 0.28
Fumaric	0.10 ± 0.00	0.15 ± 0.01	0.24 ± 0.01	0.32 ± 0.03	0.06 ± 0.00	0.69 ± 0.06	0.94 ± 0.05	1.15 ± 0.10	0.15 ± 0.01	0.21 ± 0.02	0.26 ± 0.00	0.24 ± 0.02	0.34 ± 0.02	0.43 ± 0.02	0.22 ± 0.02	0.15 ± 0.01	0.32 ± 0.01	0.20 ± 0.00	0.24 ± 0.00
Total phenolic content (µg/g)	11.30 ± 1.35	17.16 ± 1.35	17.12 ± 1.78	31.12 ± 2.43	24.53 ± 1.17	57.56 ± 7.00	62.58 ± 6.02	100.50 ± 6.07	59.43 ± 6.64	72.98 ± 4.09	96.75 ± 5.83	97.81 ± 10.52	84.55 ± 3.76	55.84 ± 3.14	85.66 ± 10.94	71.88 ± 4.18	79.61 ± 6.62	118.67 ± 10.13	120.95 ± 12.32

* Mean ± SD. n.d = not detected.

**Table 3 foods-12-03528-t003:** Mineral contents of pooled sugar sands (mean ppm ± sd).

Mineral	Low Sugar Content	Intermediate Sugar	High Sugar Content
Calcium	113,577 ± 9121	66,329 ± 6401	55,451 ± 4090
Iron	18.60 ± 3.37	16.73 ± 2.56	18.13 ± 0.75
Potassium	574 ± 17	1180 ± 98	2266 ± 115
Magnesium	3098 ± 186	1877 ± 185	3243 ± 45
Manganese	813 ± 70	676 ± 145	583 ± 68
Sodium	335 ± 140	173 ± 39	4467 ± 202
Phosphorus	2345 ± 131	1142 ± 459	1173 ± 30
Zinc	117 ± 8	76 ± 1	108 ± 4
Lead	0.50 ± 0.00	0.58 ± 0.07	0.50 ± 0.12

## Data Availability

The data used to support the findings of this study can be made available by the corresponding author upon request.

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
