# Peer review of "The Physicochemical Characterization and In Vitro Digestibility of Maple Sugar Sand and Downgraded Maple Syrups"

_foods, 2023, doi:10.3390/foods12193528_

Round 1

Reviewer 1 Report

The paper of Decabooter et al. “Physicochemical characterization and in vitro digestibility of maple sugar sand and downgraded maple syrups” aimed to study physicochemical characteristics of maple syrup by-products and investigation their digestibility using the TIM-1 digestive tract dynamic simulator. Generally, paper is well written and includes new scientific information.

Highlights and strengths of the manuscript are:

The results may further increase interest in maple syrup by-products and help develop new strategies for their alimentary application.  

Specific comments and suggested revisions:

- Sample description is fundamentally incomplete and needs additional information.

- Is there a need for figure 1? Especially since the diagram does not give an idea of the real structure of this machine. Do you have a permission to use this figure in your paper? Does authors of [Dickinson, P. A., et al. (2012). An Investigation into the Utility of a Multi-compartmental, Dynamic, System of the Upper Gastrointestinal Tract to Support Formulation Development and Establish Bioequivalence of Poorly Soluble Drugs. The AAPS Journal, 14(2), 196–205. doi:10.1208/s12248-012-9333-x] know about this?

- It is unclear what is “bioaccessibility exceeded 100%” means. You have to explain in paper the physical (or chemical) meaning of this sentence.

- Can’t find supplementary file.

With all due respect to authors, I see no possibility to recommend paperin present form until the problems were resolved.

Reviewer 2 Report

The article entitled: “Physicochemical characterization and in vitro digestibility of maple sugar sand and downgraded maple syrups” refers to the composition of different type of maple syrups, rich in sugars and with different proprieties of the polyphenols and organic acids. Due to these reasons, maple syrups hold potential as a source of polyphenols and as a sugar resource for industrial fermentation processes. Furthermore, this article stands out for its innovation in both subject matter and the experimental analyses employed. It also takes into account various aspects related to the recycling of maple syrup, including its potential as a by-product.

However, there are some recommendations for improvement: 

·      The structure of the abstract should be revised for a more conventional format, enhancing the article's overall comprehension.

·      Table 1 requires a more comprehensive caption. Parameters such as a, b, c, and L should be explicitly defined within the caption rather than in the text.

·      The data in Table 1 should be reported to the same decimal place.

·      Starting from line 214, the article references results that involve a color match. It is advisable for the authors to provide a more detailed explanation of this correspondence to enhance reader understanding.

Reviewer 3 Report

The article deals with a comprehensive physicochemical characterization of downgraded maple syrups and maple sugar sand, compared with market-grade syrup, and the respecitve in vitro digestibility.

The subject is important, the study is very well conceived, and the results are interesting.

Few comments deserve authors' attention:

- Line 326: "3.6 Evaluation digestibility of maple syrup by-products". Change to: "3.6 Evaluation of digestibility of maple syrup by-products" 

- Figure 3: Are statistical significance of the differences available for any chart? These would be very useful to understand real changes.

- Lines 363: "... such as microwaving, ultra-Turrax or ultrasound may be necessary...". None of these methods has ever shown feasibility at relevant scales. Hydrodynamic cavitation might be more suitable, see for example real-scale extraction of carbohydrates, lipids and polyphenols from almonds, which was recently published in this journal (doi:10.3390/FOODS12050935). 

- Lines 368-369: "... the inadequacy of the analytical 368 extraction step". See previous comment.

Round 2

Reviewer 1 Report

The authors taking into account the observations made by the reviewer. Considering the explanation provided by the authors the paper after correction may accepted in present form.